# Development of an Oriented Co-Culture System Using 3D Scaffolds Fabricated via Non-Linear Lithography

**DOI:** 10.3390/ma15124349

**Published:** 2022-06-20

**Authors:** Antonis Kordas, Phanee Manganas, Alexandros Selimis, Georgios D. Barmparis, Maria Farsari, Anthi Ranella

**Affiliations:** 1Institute of Electronic Structure and Laser, Foundation for Research and Technology-Hellas (FORTH/IESL), GR-71003 Heraklion, Greece; antkor89@iesl.forth.gr (A.K.); pmangana@iesl.forth.gr (P.M.); selimis@iesl.forth.gr (A.S.); mfarsari@iesl.forth.gr (M.F.); 2Department of Materials Science and Technology, University of Crete, GR-71003 Heraklion, Greece; 3Institute of Theoretical and Computational Physics, Department of Physics, University of Crete, GR-71003 Heraklion, Greece; barmparis@physics.uoc.gr

**Keywords:** 3D scaffolds, tissue engineering, surface topography, multi-photon lithography, Schwann cells, neural co-culture system

## Abstract

Damage in the Peripheral Nervous System (PNS) is related to numerous neurodegenerative diseases and has consequently drawn the attention of Tissue Engineering (TE), which is considered a promising alternative to already established methods such as surgery and autografts. TE focuses on the design, optimization, and use of scaffolds in vitro and in vivo. In this work, the authors used a novel scaffold geometry fabricated via Multiphoton Lithography (MPL), a commonly used fabrication method, for the mono- and co-cultures of glial Schwann (SW10) and neuronal Neuro-2a (N2a) cells. Both cell types have already been used for the study of various neurodegenerative diseases. However, their focus has been on only one of the cell types at a time, with studies regarding their co-culture only recently documented. Here, the suitability of the fabricated scaffolds has been explored and the effects of topography on SW10 and N2a behavior have been investigated. Our findings demonstrate that scaffold co-culture systems favor the presence of neurites compared to mono-cultures at 21 days (31.4 ± 5.5% and 15.4 ± 5.4%, respectively), while there is also a significant decrease in long neurites in the mono-culture over time (45.3 ± 15.9% at 7 days versus 15.4 ± 5.4% at 21 days). It has been shown that the scaffolds can successfully manipulate cell growth, elongation, and morphology, and these results can form a basis for the development of an experimental model for the study of PNS-related diseases and understanding of key cell functions such as myelination.

## 1. Introduction

Damage to tissues and/or organs due to trauma, disease, or congenital defects affects millions of people on a global scale. If not properly treated, damage can lead to tissue/organ malfunction or even loss of function, severely impairing the lives of patients. Tissue engineering (TE) is an emerging research field that aims to restore damaged tissues by utilizing scaffolds to act as platforms for cell cultures [1]. The ultimate goal of TE is to replace dysfunctional tissues and/or organs on demand using cells originating from each individual patient [2]. It is not surprising that such potential has generated tremendous interest in the field of TE and extensive research is ongoing, as TE seems to be a promising alternative to already-established, tried-and-tested methods, such as surgery and autografts, which have only yielded moderate success to date [3].

The natural, 3D environment of cells is essential in maintaining physiological function. This environment, the Extracellular Matrix (ECM), provides all the necessary mechanical, chemical, and biological stimuli for cells to interact and form tissues and organs. Cell behavior and function are dictated by the ECM, which provides the environment for key cellular responses, such as adhesion, orientation, proliferation, and differentiation [4,5,6]. Consequently, when developing scaffolds for TE, the aforementioned parameters should be taken into account, as mimicking the natural environment of cells can make the difference between success and failure in scaffold development [7]. Scaffold mechanical properties, surface chemistry, geometry, and micro/nano topography are all aspects that require careful consideration before attempting any experimentation or treatment and are closely related to the material used for the scaffold fabrication [8]. Materials used for TE applications should exhibit several desirable characteristics, such as biocompatibility, biodegradability (or lack of it, depending on the application), and lack of cytotoxicity, as well as mechanical stability and robustness [9].

The nervous system has drawn significant attention in the field of TE [10,11]. Typically, neural tissue consists of two major cell types: neurons and glial cells. Neurons are responsible for signal transmission and are essential for the normal function of tissues and organs, whereas glial cells play a more supportive role, by providing mechanical stability and facilitating nutrient transport. In the peripheral nervous system (PNS), glial cells are known as Schwann cells and play essential roles in tissue regeneration during injury, the transport of crucial nutrients and factors, and the mechanical support of neuronal cells through the expression of myelin. This protein creates a cover known as the myelin sheath around neuronal axons, ensuring normal neuron function, insulation, and signal transmission. Damage in the myelin sheath is associated with neurodegenerative diseases, and an inability to repair the damage can cause serious problems for patients [12]. In the PNS, nerves can spontaneously regenerate when the extent of the damage is small, while for more damaged areas, an autograft is required to bridge the gap created by the severed axon. As such, TE for the repair of PNS injuries focuses on creating appropriate cellular environments that facilitate regeneration and guidance [13].

Various research studies have focused on these two cell types individually. However, as they are closely related in vivo, studies using both cell types simultaneously in co-culturing systems could provide great insight into modelling related diseases and injuries. These are relatively limited but have shown that when designing 3D scaffolds for PNS applications, researchers must carefully account for scaffold characteristics, such as topographical features and chemical and physical properties, as they can all affect cellular responses and behavior. Thus, the choice of the suitable scaffold is both dependent on and plays a key role in the application it is designed for [10].

Several methods for scaffold fabrication have been developed over the years. Among them, Multiphoton Lithography (MPL) stands out as it allows the fabrication of large 3D scaffolds with a high (submicron) resolution [14,15]. MPL is a Direct Laser Writing (DLW) technique that uses femtosecond (fs) laser pulses for the polymerization of photosensitive materials. With the aid of Computer-Aided Designs (CADs), the user is able to fabricate complex designs of various sizes, shapes, and topographies while maintaining a high resolution, which, in turn, can provide submicron characteristics that are essential for specific TE applications. Oftentimes, the topographical characteristics play a vital role in cell responses such as cell migration and morphology, a fact that makes MPL an invaluable fabrication method for TE scaffolds [16].

A common drawback of MPL is that fabricated scaffolds exhibit autofluorescence due to the use of photoinitiators (PIs). This might not be a consideration when studies utilize techniques such as Optical Microscopy or Scanning Electron Microscopy (SEM); however, it is a major disadvantage when techniques such as Fluorescence/Confocal Microscopy are used. This obstacle can be overcome by using materials with PIs that exhibit no autofluorescence and are commercially available. This solution is not always optimal, as these materials may be difficult to handle, yield low-quality scaffolds, exhibit non-desirable properties, and may not be cost-effective. Recently, Flamourakis et al. [17] proposed a protocol that utilizes Sudan Black B (SBB) as a PI, as well as a post-fabrication treatment of an already-established hybrid (organic/inorganic) material [18] to minimize scaffold autofluorescence. This approach provides an affordable alternative, which shows promising results by using an already-established material that is cheap, simple to prepare, easy to handle, yields high-quality structures with excellent resolution, and exhibits good biocompatibility.

In this work, the fabrication of scaffolds with minimal autofluorescence for the cultivation of neurons and Schwann cells, both in mono- and co-culture systems, for three different timepoints (7, 14, and 21 days) was carried out. A novel scaffold geometry was fabricated, and the elongation of the cells and neurites in a direction of choice was showcased. By using two imaging techniques (Scanning Electron Microscopy (SEM) and Confocal Microscopy), it is shown that long neurite growth (>40 μm) is favored for scaffold co-culture systems for long-term experimental periods compared to mono-culture setups where long neurites exhibit a significant decrease from 7 to 21 days. These findings highlight that, by choosing scaffolds with appropriate geometry, cell behavior, directionality, and axon growth can be influenced according to application needs. The presented results provide the first step in the creation of an in vitro model for studying neuronal cell behavior, while also providing insight into the understanding of neurodegenerative diseases and cell processes, such as myelination, in future research.

## 2. Materials and Methods

### 2.1. Photoresist Preparation

The material used in this study was prepared as previously described [17]. Briefly, glass coverslips were lyophilized by immersion in 93% ethanol and sonicated for 1 h. The ethanol was removed, and a mixture of methacryloxypropyl trimethoxysilane (MAPTMS, 99%, Polysciences Inc., Warrington, PA, USA)/dichloromethane (ratio 80:1) was added, followed by a 4 h sonication step. The mixture was then substituted with 70% ethanol and another sonication step of 1 h followed. Lastly, the ethanol was discarded and replaced with fresh 70% ethanol for storage. The whole procedure was conducted at room temperature (RT).

The material used is the already reported SZ2080 (organic–inorganic hybrid) [18]. MAPTMS (99%, Polysciences Inc., Warrington, PA, USA) was hydrolyzed with 0.1 M HCl, while at the same time, methacrylic acid (MAA, 98%, Sigma-Aldrich, St Louis, MO, USA) was combined with zirconium n-propoxide (ZPO, 70% in propanol, Sigma-Aldrich, St Louis, MO, USA) at a molar ratio of 1:1 and stirred for 30 min in a separate container. The contents of the two containers were then slowly combined, and after stirring for another 30 min, the photoinitiator was added to the mix. Two different photoinitiators were used: (a) 4,4′- bis (diethylamino) benzophenone (Michler’s ketone, Sigma-Aldrich, St Louis, MO, USA), which was added directly to the mixture at a concentration of 1% proportionally to the monomers, and (b) Sudan Black B (SBB, Sigma-Aldrich, St Louis, MO, USA), which was firstly diluted in isopropanol and through a second dilution was added to the mix at a final concentration of 0.04% proportionally to the monomers. The mixtures were further stirred for approximately 20 min and then filtered using a 0.2 μm pore size filter. The lyophilized coverslips were dried, and a droplet (15 μL) of material was deposited in the center. The droplets were then heated at 85 °C for 2 ½ h, to remove any excess moisture and to make the material obtain a gel-like composition. The procedure was conducted at RT and in the absence of light.

### 2.2. Bridge-Shaped 3D Scaffold Fabrication via MPL

The scaffolds fabricated for this study were pyramid-shaped bridges (Figure 1A,B). This geometry consists of pairs of blocks that act as support for guidewires that connect them. The guidewires are of varying lengths and are expected to facilitate the selective orientation of cells. The dimensions of the scaffolds were 400 μm × 400 μm × 60 μm, which are considered large enough to support cell cultures up to 3 weeks (21 days).

A femtosecond fiber laser (FemtoFiber pro NIR, Toptica Photonics AG, Munich, Germany) operating at 780 nm was used as an irradiation source (pulse duration: 120 fs, repetition rate 80 MHz) to fabricate the scaffolds for the experiments.

Fabrication was achieved using a 20× (NA = 0.8, Zeiss, Plan Apochromat) microscope objective lens. The scanning speed was 50 mms^−1^, the power output was 40 mW, the distance between scaffold layers was 6 μm (10 layers in total), and the hatching was set at 0.5 μm. The scaffolds prepared with Michler’s ketone were used for experiments visualized via SEM, while the ones prepared with SBB were used for experiments visualized via Confocal Microscopy.

After fabrication, the samples were developed in 4- methyl- 2- pentanone/ 2- propanol (ratio 1:1) until all unpolymerized material was removed (typically 15 min). The coverslips were then transferred to a solution of 2- propanol/ d-H_2_O (ratio 1:1) for 2 min and were left to dry. They were then placed in a 24-well culture plate for storage. Prior to cell experiments, the scaffolds were immersed in 0.3% SBB in 70% ethanol as described in [19] for 2 h in order to eliminate any remaining autofluorescence [20]. Subsequently, multiple washes with 70% ethanol removed any excess SBB. All scaffolds were sterilized before use in cell cultures, where they were immersed in 70% ethanol for 10 s, left to dry, and then sterilized under UV light for 30 min, alongside flat glass coverslips that serve as controls.

### 2.3. Thin Film Preparation and Contact Angle Measurements

Thin films of both resins were prepared for contact angle measurements. The preparation of thin films is similar to the preparation of droplets as described above. More specifically, 15 μL of the material was deposited in the center of a coverslip, which was spin-coated at 4000 rpm for 40 s with an acceleration step of 500 rpm/s. The thin films were then heated for 2 ½ hours at 85 °C to remove any excess moisture. Afterward, they were placed under a UV light overnight to polymerize the resins. The next day, the films were immersed in 4-methyl-2-pentanone/2-propanol (ratio 1:1) for 15 min followed by immersion in 2-propanol/d-H_2_O (ratio 1:1) for 2 min to remove any unpolymerized material and were left to dry.

Contact angle measurements were made in triplicate using a contact angle computing device (OCA 15 Plus, Data Physics Instruments, Filderstadt, Germany). The samples tested were (a) material with Michler’s ketone as a PI (developed), (b) material with SBB as a PI (developed), and (c) material with SBB as a PI (undeveloped). A 2 μL droplet of dd-H_2_O was placed in three different areas of each film to obtain an average contact angle of the film (Figure 1C). The average values and standard deviations of all films of all three cases were then calculated with an accuracy of 2 decimals.

### 2.4. Cell Seeding and Cultures on Flat Controls and 3D Scaffolds

Murine Schwann (SW10) and Neuro-2a (N2a) cell lines (used between passages 1–5) were used for both mono-cultures and co-cultures on the fabricated scaffolds for 3 different timepoints (7, 14, and 21 days). To avoid overpopulation on the scaffolds due to the extended periods of culture, a starting population of 5 × 10^3^ for SW10 cells and 10^4^ for N2a cells were used. The desired number of cells was seeded on the flat glass coverslips in a total volume of 1 mL, whereas, for the scaffolds, a volume of 100 μL was placed directly above the scaffolds and left for 10 min to ensure the cells would be near the scaffold region, before adding the remaining 900 μL of the medium. The medium used was Dulbecco’s Modified Eagle’s Medium (DMEM, Gibco, Grand Island, NY, USA), supplemented with 10% Fetal Bovine Serum (FBS, Gibco, Grand Island, NY, USA) and 1% antibiotic (penicillin/streptomycin, Gibco, Grand Island, NY, USA). The cells were then incubated at 37 °C in a 5% CO_2_ atmosphere. The following day, the medium was removed, and the cells were supplied with a differentiation medium (DMEM supplemented with 0.5% FBS, 1% penicillin/streptomycin, and 0.3 mM cyclic adenosine monophosphate (cAMP) (Sigma-Aldrich, St Louis, MO, USA). A fresh differentiation medium was provided every 2–3 days until the timepoints were reached.

### 2.5. Scanning Electron Microscopy Imaging

At each timepoint, the medium was removed, followed by two 7 min washes with 1 × PBS on ice. Samples were subsequently fixed with 2.5% glutaraldehyde (GDA, Sigma-Aldrich, St Louis, MO, USA) in 0.1 M Sodium Cacodylate Buffer (SCB) for 45 min and two 7 min washes with 0.1 M SCB were performed. Dehydration was achieved with 7 min immersions of the samples in ethanol gradients (30%, 50%, 70%, 90%, 100%). Samples were dried with 100% dry ethanol through a Critical Point Drying (CPD) process using a Baltec CPD 030 instrument (BAL-TEC AG, Balzers, Liechtenstein). Prior to SEM imaging, all samples were sputter-coated for 78 s each (approximately 20 nm Au thickness) using a Baltec SCD 050 instrument (BAL-TEC AG, Balzers, Liechtenstein).

### 2.6. Immunofluorescence Staining

At each timepoint, the medium was removed followed by three washes with 1xPBS. The cells were fixed using 4% *w*/*v* paraformaldehyde (PFA, Sigma-Aldrich, St Louis, MO, USA) in 1 × PBS for 15 min, permeabilized with 0.2% *v*/*v* Triton X-100 (Sigma-Aldrich, St Louis, MO, USA) in 1 × PBS for 10 min, and blocked with 2% *w*/*v* Bovine Serum Albumin (BSA, Biofroxx, Einhausen, Germany) in 1 × PBS for 1 h at RT. Cells were stained in 0.5% *w*/*v* BSA in 1 × PBS with a-Tuj1 (BioLegend, San Diego, CA, USA), TRITC-conjugated phalloidin (EMD, Millipore, Burlington, MA, USA), or phalloidin CF-680R (Biotium, Fremont, CA, USA) (all 1:1000) for 1 h. a-Synaptophysin (Abcam, Cambridge, UK) was also used at 1:1000 dilution o/n at 4 °C. Secondary CF-488A rabbit a-mouse and CF-555 goat a-rabbit antibodies (Biotium, Fremont, CA, USA) were used (both 1:500) for 1 h at RT. Nuclei were counterstained with DAPI (Life Technologies, Carlsbad, CA, USA) and mounted on microscope glass slides. Sample observation and imaging were performed using an Inverted Confocal Microscope (Leica SP8 Inverted Confocal, Leica Microsystems, Wetzlar, Germany). All image processing was conducted with the open-source Fiji ImageJ program. Table 1 summarizes the experimental conditions used for SW10 mono-cultures, N2a mono-cultures, and SW10/N2a co-cultures used in this work.

### 2.7. N2a Differentiation and Neurite Length

N2a mono- and SW10/N2a co-cultures were evaluated at all timepoints by counting all N2a differentiated cells (Tuj1 positive) (all experiments were performed at least in triplicate). Any N2a cell that exhibited one or more neurites, i.e., axons and/or dendrites as described in [21], was considered differentiated; however, only the longest neurite for each cell was accounted for [22]. In the case of scaffold experiments, only N2a cells that were in direct contact and N2a cells that were not in direct contact but had neurites that were in direct contact with the scaffold guidewires were taken into account. An area comparable to a scaffold in size was measured in flat controls to maintain uniformity in both cases.

For the purposes of this work, the neurites were divided into two categories: (1) Short neurites up to 40 μm long (approximately twice the length of a typical N2a cell) and (2) long neurites over 40 μm long, which consisted exclusively of axons. Neurite length was calculated manually by measuring the distance between the protrusion from the N2a body to the end of the neurite with an accuracy of 2 decimals.

## 3. Results

The main findings of this work are presented in this section. 3D scaffolds were fabricated, and mono-cultures of both SW10 and N2a cells, as well as co-cultures, were carried out. Cell behavior between flat controls and scaffolds was monitored in order to determine how the scaffold may be able to influence cellular parameters such as morphology, directionality, differentiation, and neurite formation. To this end, both scanning electron microscopy and confocal microscopy were used.

### 3.1. Scaffold Fabrication

Pyramid-shaped scaffolds (Figure 1A,B) were fabricated via MPL. The dimensions of 400 μm × 400 μm × 60 μm were chosen, as they provide large enough scaffolds to sustain long-term cultures without being significantly damaged by large numbers of cells or frequent medium changes, while also maintaining satisfactory resolution for the purposes of this work.

The experimental setup (Galvo system) has already been described in detail elsewhere [23]. In this setup, galvanometric mirrors move the laser beam on the x and y planes, whereas movement on the z-axis is achieved via linear stages. This setup can produce large and bulky structures in a time-effective manner where structure resolution is only a secondary concern. To highlight the efficiency of the system, each individual scaffold needed approximately 2 ½ min to be completely fabricated. Six individual scaffolds were fabricated per coverslip adding up to a final fabrication time of approximately 15 min.

The aim of this work is to investigate the presence of guidewires. as well as their length, have a significant impact on cell directionality, morphology, and neurite outgrowth.

### 3.2. Contact Angle Measurements

Contact angle measurements were conducted in triplicate using thin films, as described in Section 2.3. First, 2 μL droplets were deposited in three random areas of each film, enabling the calculation of an average contact angle of each film. The average contact angles and their standard deviations were calculated. The results are shown in Figure 1C and summarized in Table 2.

As can be seen above, the contact angles in all cases are below 90°, which is typically the limit to consider a material hydrophilic or hydrophobic. As such, the materials in all three cases should exhibit hydrophilic properties, which are typically regarded as beneficial for cell adhesion. The work of Flamourakis et al. [17] showed that post-fabrication, SBB treatment smoothens the scaffold surfaces. Here, the approach was different; instead of a post-fabrication treatment, both SBB-Developed and SBB-Undeveloped films were used. Development was expected to remove unpolymerized resin and thus affect the uniformity of the films. Undeveloped films, on the other hand, had a much more uniform surface. The effect of the development step is immediately observable, as the contact angle shows an approximate 10° increase compared to the other two cases. The effect of the post-fabrication treatment with SBB reported in [17] could be compared to the effect of omitting the development step here; both approaches give smoother surfaces, which should exhibit hydrophilic properties.

It should be mentioned that no undeveloped films were used in the case of Michler’s ketone PI. Considering the fact that scaffolds prepared with this PI are suitable only for SEM imaging, an SBB post-fabrication step was not necessary and thus was not used. This, in turn, means that the scaffolds fabricated with this resin should have a closer profile to the films used, making the examination of undeveloped Michler’s ketone films redundant for the purposes of this study. The second consideration is that contact angle measurements were not made for scaffolds. Given the relatively small scaffold size and the limitations of our system to fabricate larger non-periodic scaffolds, it was not possible to carry out such measurements. However, polymerized films should resemble the properties of the 3D scaffolds, and as such, exhibit hydrophilic properties, enabling easier cell adhesion during cell cultures.

### 3.3. Cell Cultures

Cell cultures were conducted at least in triplicate alongside their respective control samples that were performed on 13 mm glass coverslips. Cellular orientation and morphological differences between glass surfaces and scaffolds were investigated. In order to examine whether a co-culture system would have differences compared to cultures with a single cell type, mono-cultures of both SW10 and N2a cells were also carried out. In all cases, the cells were observed using both SEM and confocal imaging at three timepoints (7, 14, and 21 days).

#### 3.3.1. SW10 Mono-Cultures

In the case of SW10 mono-cultures, SW10 cells exhibited a random orientation on glass coverslips (Figure 2). As expected, greater saturation of the surface was observed over time. In Figure 2, it can be seen that the morphology of the cells is quite flat, while their orientation is random and follows no specific pattern, which is also complemented by the confocal images in Figure 2B. Actin cytoskeleton staining shows that the cells spread in all directions at all three timepoints.

On the other hand, when SW10 cells are cultured on scaffolds (Figure 2C,D), especially on or near the guidewires, they tend to elongate along the wire axis. As expected, due to the low initial seeding density, fewer cells are attached to the scaffolds at the 7-day timepoint, and one can see that their orientation is not significantly affected when compared to later stages (14 and 21 days). As days pass, more cells migrate onto the scaffolds from the surrounding area and proliferate. This is demonstrated in Figure 2C (21 days), where cells can be seen ‘’climbing’’ the supporting blocks of the structures, which leads to greater scaffold coverage. Here, one can observe that cell directionality is dictated by their environment, which, in this case, is the scaffold guidewires, while the cells that are on the supporting blocks and the area surrounding the scaffolds mostly behave similarly to the glass controls. Furthermore, it seems that the wire length does not appear to play a significant role in cell directionality, even though it is obvious that more cells attach to the longer wires. It is worth noting that the guidewires are the only stimuli provided to the cells, indicating that even a simple geometry feature can be enough to affect important cell characteristics.

#### 3.3.2. N2a Mono-Cultures

N2a mono-cultures were performed in a similar manner as described above. It was expected that, since SW10 elongation can be guided by scaffold geometry, axon elongation could be affected in a similar manner. To this end, the cells were cultured in a differentiation medium containing cAMP, and the visualization of Tuj1 expression (which stains for tubulin beta-III, a neural cell-specific marker) was performed to obtain more information regarding axon growth.

Some axon growth can be observed as early as the first timepoint (7 days) and is visible both on the glass slides (Figure 3A,B) and on the scaffolds (Figure 3C,D). In some cases, N2a cells interact between them (Figure 3A, 14 days), although, similarly to the SW10 cells, the orientation of neurite growth is completely random.

When the N2a cells were grown on the scaffolds, they exhibited directed axon growth along the scaffold guidewires. However, not all cells exhibited axon growth, even though the topography offers the possibility of selective orientation. In addition, cells that attached to the supporting blocks showed mostly random axon growth.

However, when observing the images obtained after 21 days (Figure 3C,D), even though scaffold saturation is obvious, fewer axons were observed along the guidewires. This phenomenon is not clearly understood but could possibly be attributed to cell competition due to scarcity of the differentiation factor and/or a large number of cells and space limitations. These findings indicate that while 3D topography can indeed influence axon growth to some degree, there is the potential for the development of a more efficient protocol in the near future.

#### 3.3.3. SW10/N2a Co-Cultures

In addition to the mono-cultures shown above, co-cultures with both cell types were also performed. The important question was whether these cells could co-exist on the scaffolds and if their behavior would change compared to the mono-culture systems. In this case, quadruplicate staining was performed for confocal imaging, with the addition of synaptophysin to examine whether the N2a cells remain functional in this culturing system.

In Figure 4, it can be seen that when co-culturing both cell types on glass coverslips, a similar behavior to what has previously been reported [24] is observable. More specifically, SW10 cells tend to attach to the glass surface, while N2a cells either attach to the glass (if there is empty space) or lay on top of the SW10 cell layer. The latter is expected, as the physiological role of SW10 cells is to provide support for the neurons. Morphologically, neither cell type appears to be significantly affected by the presence of the other compared to the mono-cultures. A portion of the N2a cells also grow axons that follow random directions as shown above (Figure 4A).

Similar information can be obtained from the respective images in Figure 4B. SW10 cells are only stained with two dyes, DAPI and phalloidin (blue and magenta, respectively), whereas N2a cells are stained with all four dyes. N2a cells appear both on the glass and on top of SW10 cells, and axon growth is again observed, highlighting the behavioral consistency of both cell types in flat co-culture systems.

As expected, the co-culture scaffold system was more challenging than the individual mono-cultures. Indeed, even though scaffold saturation is progressively observed with the passing of days, and although both cell types appear on the scaffolds, there are some key points that need to be mentioned. Firstly, it can be seen that the SW10 cells almost exclusively attach to the scaffold and saturate it, while the N2a cells appear to attach to the SW10 cells (Figure 4C,D). This is expected, given the results shown in Figure 4A,B. However, fewer neurites can be observed on the scaffolds compared to the flat controls. This might not be surprising since only neurites that come in contact with the scaffold guidewires are investigated. This spatial limitation means that fewer cells are regarded compared to a similar flat area and, consequently, fewer neurites are taken into account. This change may also be correlated to the large number of SW10 cells, which cover the scaffolds almost completely. The possibility that N2a cells need to come in more direct contact with the scaffold regardless of what happens physiologically in vivo cannot be ruled out. As previously stated, a competition event due to the scarcity of the differentiation factor is still a possibility. Furthermore, other causes that are not yet obvious or understood may also play a role in the reduction of axon growth. In the cases where axons form on the scaffold guidewires, it can be observed that they follow the direction provided by the wire, as previously demonstrated (Figure 3C,D). Comparatively, SW10 behavior does not seem to change significantly when compared to the mono-cultures (Figure 2). This could potentially be attributed to the fact that SW10 cells attach to the scaffolds first and cover it exactly as they would do in mono-cultures, whereas N2a cells migrate on the scaffolds at a later point and have more difficulty adapting. In either case, one can observe that the topography can influence cell morphology, orientation, and directionality.

#### 3.3.4. N2a Differentiation

The percentage of short and long neurites of differentiated N2a cells was calculated in all cases where N2a cells were present. As described in Section 2.6, neurites were divided into two categories: Short (up to 40 μm) and long (above 40 μm). In the latter case, neurites were comprised exclusively of axons, whereas in the former case, both axons and a small number of dendrites were also used in the calculations.

An investigation of whether the different conditions (mono-cultures vs. co-cultures, scaffold presence, different timepoints) would show any preference towards a specific neurite group (short or long) was carried out. The mean values and standard deviations of each group were calculated in an attempt to determine whether specific lengths are favored in specific experimental conditions.

Figure 5 shows the percentage of short and long neurites for N2a mono-cultures (Figure 5A) and SW10/N2a co-cultures (Figure 5B). It is obvious that, in all cases, the majority of the neurites are in the first category (short, up to 40 μm). In the case of flat controls, the percentage of both short and long neurites is stable throughout all the timepoints tested for both mono- and co-cultures, and the small differences observed lie within the experimental error. More specifically, timepoints of one condition (7 vs. 14 vs. 21 days) show no statistical significance between them (Figure 5A). This is not surprising since there is no obvious reason for a preference for specific neurite lengths just by the passing of days. Additionally, although longer axons show an increase in the co-cultures compared to their respective mono-cultures (approximately 10% increase, Figure 5B), t-tests showed that these results lie within the experimental error. This indicates that although the presence of SW10 cells slightly shifts the preference towards longer axons, this stimulus alone is not enough to have a noteworthy impact on that direction.

As far as the scaffolds are concerned, it is obvious that there is a more dramatic decrease in axons with the passing of days (Figure 5A). For long-term mono-cultures, this decrease is statistically significant when comparing 7-day and 21-day cultures (long axon percentages of 45.3 ± 15.9% and 15.4 ± 5.4%, respectively, unpaired t-test: *p*-value = 0.037) (Figure 5A). This is not the case for the co-culture system (Figure 5B). In the case of co-cultures, the percentage of long axons remains practically stable at all timepoints. However, when comparing the 21-day mono-cultures and co-cultures on scaffolds, it is obvious that the increase in the percentage of long neurites is statistically significant (long axon percentage of 21-day co-cultures on scaffolds: 31.4 ± 5.5%, unpaired t-test: *p*-value = 0.0233). This observation is particularly interesting, as the only difference between the two cases is the presence of SW10 cells. While the reasons for this decrease in mono-cultures are unknown, the results indicate that the presence of SW10 cells seems to facilitate the growth of longer axons over time.

## 4. Discussion

Tissue Engineering has emerged as a field to counter problems derived from damaged tissues due to injury or disease. Specifically, in the PNS, the classic methods that are used to repair damage due to injury or disease are autografts and surgery. However, their success is not always guaranteed, which has given room for TE to be considered as an alternative way of dealing with this problem. In this work, an attempt to develop a system as a basis for the study of such diseases has been made. An established material has been used for the fabrication of scaffolds of specific geometry and dimensions, which served as the environment of SW10 and N2a mono-cultures, as well as SW10/N2a co-cultures.

The material used has already been established as cytocompatible in previous works in our lab [25,26]. Moreover, it has been demonstrated that by using SBB as a PI [17], autofluorescence can be drastically reduced. An alternative would be the use of non-fluorescent PIs such as Irgacure 369; however, to our knowledge, even these PIs exhibit fluorescence after polymerization, which is higher compared to that of SBB. Softer materials such as hydrogels can be of interest to TE regarding the PNS; however, scaffolds fabricated with such materials can have low resolution. In our case, a novel geometry consisting of blocks supporting guidewires of different lengths was used. The aim was to test two important factors: (a) To examine if such scaffolds could provide a favorable environment for SW10 and N2a mono-cultures and co-cultures and (b) to examine if guidewire length could affect cell behavior. It was found that guidewire length does not seem to be a limiting factor, although fewer cells can fit on the shorter wires. Cells on the wires tend to follow the geometry, regarding both SW10 elongation and N2a axon growth, while cells that are on the supporting blocks, on the sides of the blocks, in the surrounding area of the scaffolds, and in the flat areas of the coverslips tend to follow a random orientation. This shows that topography alone can influence cell behavior to an extent. In neural tissue engineering of the PNS, this is essential, as conduits that enable axon elongation in specific directions are key for successful tissue repair [27,28].

Another key question was what would happen when both cell types were cultivated together compared to their respective mono-cultures. With the help of SEM and confocal microscopy, information about cell morphology and important molecules that are present in synapses during cell attachment was gathered. Cell behavior was not expected to drastically change in the co-culture system compared to the mono-cultures. In all three cases, cells progressively saturate their environment (flat glass or scaffold), with scaffold saturation occurring due to cell migration over time. Scaffold topography was able to influence cell morphology and axon orientation. An interesting observation is that in the co-culture system, both on flat coverslips and scaffolds, the SW10 cells tend to be in direct contact with the environment, and N2a cells prefer to attach on top of SW10. This is not a surprise as, physiologically, SW10 cells play a supporting role.

These findings are in good agreement with previous work in our lab [24]. More specifically, Angelaki et al. have designed patterned micro-sized spikes and nano-sized ripples for the co-culture of SW10/N2a cells. There, SW10 cells showed directed adhesion on the spikes, which played a pivotal role in the direction of differentiated N2a cells. Similar to what has been presented in this work, SW10 cells saturated the scaffold surface and followed an orientation based on scaffold topography. N2a cells exhibited a similar behavior; they showed a preference for SW10-covered areas, and their neurite directionality was dramatically influenced by the SW10 orientation. In the case of flat coverslips, this orientation was random, while on the other hand, the orientation of cells and directionality of neurites on scaffolds followed the scaffold geometry. The same observation could be made for N2a mono-cultures, as neurite directionality was also influenced by the scaffolds even in the absence of SW10 cells. Angelaki et al. proposed that undifferentiated N2a cells do not sense topographical cues; the process of topography sensing seems to come from the neurites of differentiated N2a cells. This is also featured to an extent in the presented results, as in N2a mono-cultures (Figure 3), differentiated N2a cells show neurite elongation that is directly affected by scaffold guidewires, whereas the same observations are made for co-cultures with the important detail that SW10 cells are always present near or under differentiated N2a cells that exhibit neurite directionality (Figure 4). Key differences between these two works are the fact that in this work, the fabricated scaffolds are three-dimensional, whereas, in the work of Angelaki et al., the scaffolds constitute a 2.5D environment. In addition, the starting populations of cells in the current work are far smaller, as the aim of the current study was to establish the conditions for a long-term coculturing system. As such, the lower cell numbers allowed us to observe scaffold saturation progressively over time. It is very interesting that similar cell behavior is present in both works, with SW10 cells seeming to be more sensitive to topographical cues and forming a glial carpet that supports and influences neurite directionality of differentiated N2a cells that are in contact with them. Combined, these findings show that this behavior is encountered in both 2.5D and 3D environments and that it can be observed for different topographical characteristics.

Another concern that was mentioned previously was that after 21 days, fewer N2a axons were observed on the scaffold wires. This could be due to the limited availability of the differentiation factor due to greater N2a numbers. Axon retraction, degeneration, or shedding (a phenomenon known as synaptic pruning) might also be at play [29,30], but there might also be other unknown reasons for these results. It is worth noting that the comparison of the two neurite groups (short and long) showed that no significant changes were observed on flat controls, as expected; however, in the case of mono-cultures on scaffolds, this changed; longer axons were not favored, and their percentage decreased significantly. The reason this could be happening is still under investigation; however, it can be regarded as a relative hindrance since the preference for longer axons should be considered an advantage (grafts/scaffolds are designed to close large gaps of severed neurons during surgery). This phenomenon seems to be absent in the case of the co-cultures. The presence of SW10 cells is the only parameter that changes; however, it is not clear if only SW10 contributes to this observation or if there is a synergistic effect between the presence of SW10 cells and the scaffolds. It has already been reported that SW10 cells and topographical characteristics play a synergistic role for both in vivo [28] and in vitro [31,32] applications and work together to enhance neurite outgrowth, alignment, and length, as they have been used as conduits to close gaps above 20 μm [33]. Thus, it is possible that the presence of both the SW10 cells and scaffolds has a positive effect on the growth of longer axons. This means that a carefully combined co-culture system could sustain long-term cultures more efficiently without risking performance decrease. In addition, the incorporation of supporting cells such as SW10 has already shown beneficial results and it is believed that co-culture systems will gain even more attention in TE in the future. It must be noted that the findings in this work aim to offer an in vitro model for the study of neurodegenerative diseases and key cell responses such as myelination. However, the ultimate goal is to apply TE in real life. To this end, the possibility of using primary cells instead of cell lines would be a very interesting extension of this study. Th use of primary cells could pave the way for the incorporation of the presented findings in in vivo studies such as the creation of autografts for surgery or for injectable scaffolds in the near future. Furthermore, the general concept of the presented study could also be extended to other fields of TE, such as bone tissue engineering.

## 5. Conclusions

In this work, an in vitro model for the study of an SW10/N2a co-culture was developed. An easy-to-handle established material was used for the fabrication of scaffolds for cell cultures. This study showed that by carefully adjusting the scaffold topography, cell morphology and axon growth can be influenced in specific directions. Both in mono-cultures and co-cultures of SW10 and N2a cells, it was shown that cells migrate and proliferate on the scaffolds and that they are stimulated to follow the guidewire direction compared to the random orientation observed on flat glass surfaces. It is also demonstrated that the presence of SW10 cells favors the development of longer neurites on scaffolds for long-term cultures compared to mono-cultures, which could prove a valuable asset for practical applications. It is expected that by improving the presented system (larger-scale scaffolds, even further improved hydrophilicity, use of primary cells, possibility for practical applications), it could potentially serve as an even more complete tool in the fields of neurodegenerative disease study and tissue engineering.

## Figures and Tables

**Figure 1 materials-15-04349-f001:**
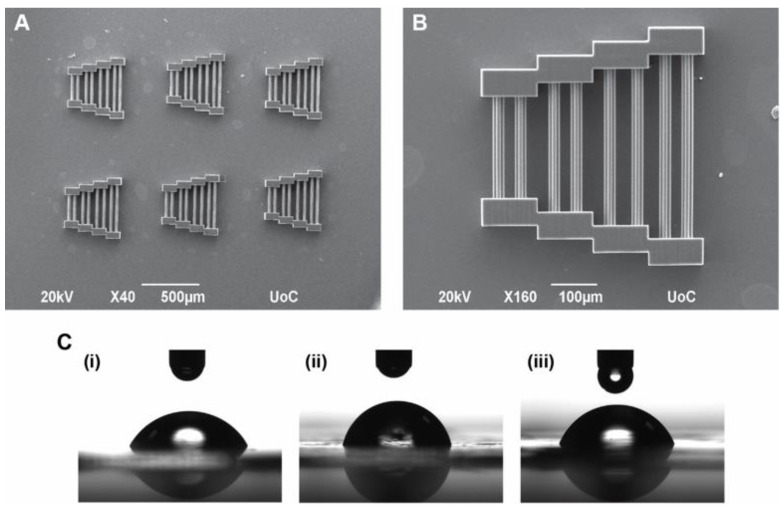
SEM image of fabricated scaffolds via MPL for cell cultures and contact angle measurements of thin films prepared from the same resins used for scaffold fabrication. (**A**) Typical coverslip with 6 scaffolds used for the experiments. (**B**) Magnification of a single scaffold. (**C**) Contact angle of thin films. (**i**) Material prepared with Michler’s ketone as a PI. (**ii**,**iii**) Developed (**ii**) and undeveloped (**iii**) films prepared with SBB as a PI. Average contact angles: 62.68 ± 1.83° (i), 75.19 ± 4.43° (**ii**), 65.87 ± 1.88° (**iii**).

**Figure 2 materials-15-04349-f002:**
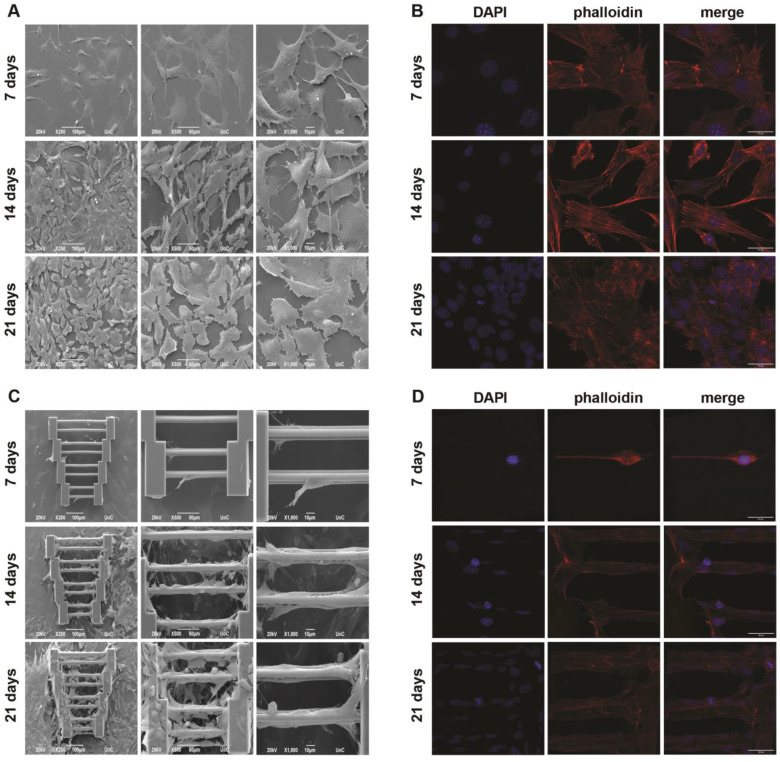
SW10 mono-cultures. (**A**,**B**) Cultures on flat glass coverslips. SEM (**A**) and confocal (**B**) images show that the cells have a random orientation. (**C**,**D**) Cultures on scaffolds. SEM (**C**) and confocal (**D**) images show a selective orientation along the guidewires, at all timepoints. Scaffold saturation does not seem to be a limiting factor for cell directionality. Scale bars in panels (**B**) and (**D**) are given in the merged images and are 30 μm.

**Figure 3 materials-15-04349-f003:**
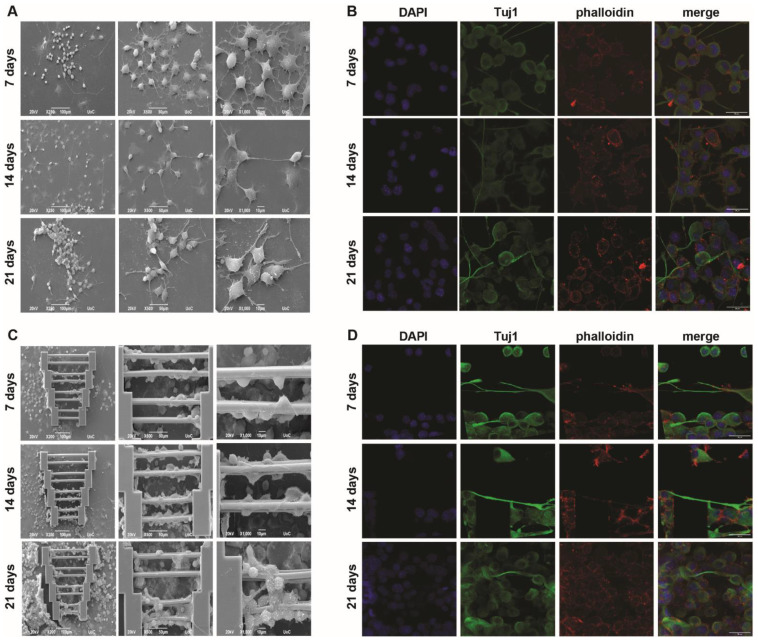
N2a mono-cultures. (**A**,**B**) Cultures on flat glass coverslips. SEM (**A**) and confocal (**B**) images show that neurites have a random orientation. In some instances, interactions of N2a cells can be observed, with random neurite orientation. (**C**,**D**) Cultures on scaffolds. SEM (**C**) and confocal (**D**) images show that cells that grow axons on the guidewires exhibit a selective directionality, which may or may not continue if the axons extend outside the guidewires. Scale bars in panels (**B**) and (**D**) are given in the merged images and are 30 μm.

**Figure 4 materials-15-04349-f004:**
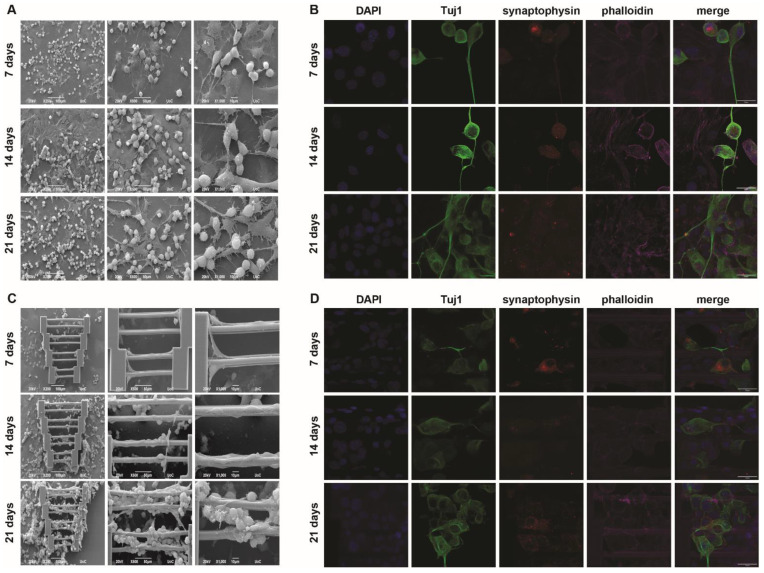
SW10/N2a co-cultures. Cultures on flat glass coverslips. SEM (**A**) and confocal (**B**) images show that SW10 cells attach to the glass substrate, whereas N2a cells attach both to the empty space of the glass substrate and on top of the SW10 cells. Some neurite growth towards random directions can also be observed. (**C**,**D**) Cultures on scaffolds. SEM (**C**) and confocal (**D**) images show that SW10 appear to attach first to the scaffolds and their behavior does not change drastically compared to mono-cultures, as they follow the guidewire orientation. N2a cells mostly attach to the SW10 cells. When N2a cells grow axons that are on the guidewires, the axons maintain guidewire directionality. Scale bars in panels (**B**,**D**) are given in the merged images and are 30 μm.

**Figure 5 materials-15-04349-f005:**
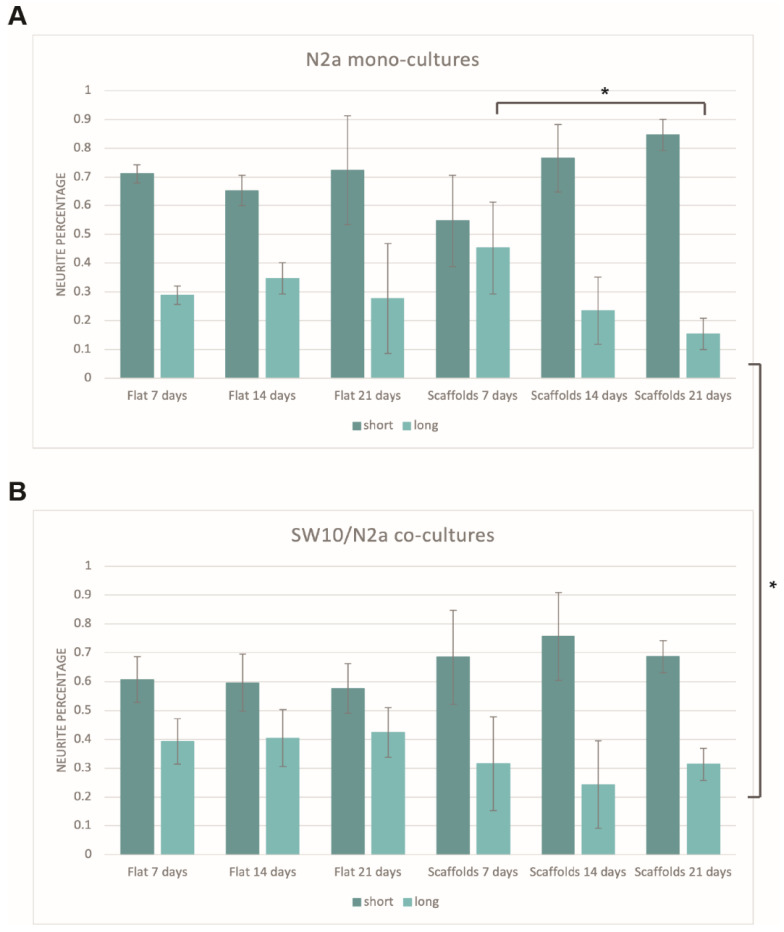
N2a differentiation. A, B: Percentage of short (dark green) and long (light green) neurites for mono- (**A**) and co- (**B**) cultures. Short neurites: Up to 40 μm, Long neurites: Above 40 μm. For flat controls, short and long neurite populations are relatively stable for both mono- and co-cultures. In the case of mono-cultures on scaffolds, long axons decrease in number over time, which is alleviated in the respective co-culture cases (*n* = 3). (* *p* < 0.05).

**Table 1 materials-15-04349-t001:** Summary of the experimental conditions and parameters used in this work. Experiments were conducted at least in triplicate for each experimental condition.

Experimental Conditions	SW10 Mono-Cultures	N2a Mono-Cultures	SW10/N2a Co-Cultures
** Starting cell population **	5 × 10^3^	10^4^	5 × 10^3^/10^4^
**Flat controls/** **scaffolds**	Yes/Yes	Yes/Yes	Yes/Yes
** All timepoints **	Yes	Yes	Yes
** SEM imaging **			
** SBB post-fabrication **	Not necessary	Not necessary	Not necessary
** Confocal imaging **			
** SBB post-fabrication **	Yes	Yes	Yes
** DAPI **	Yes	Yes	Yes
** TRITC-conjugated phalloidin **	Yes	Yes	No
** Phalloidin CF-680R **	No	No	Yes
** Tuj1 **	No	Yes	Yes
** Synaptophysin **	No	No	Yes

**Table 2 materials-15-04349-t002:** Average values and standard deviations of contact angle measurements of thin films of both resins used in this work. (MK—Michler’s ketone, SBB—Sudan Black B).

Sample	MK-Developed	SBB-Developed	SBB-Undeveloped
** Film 1 **	61.71°	76.15°	66.83°
** Film 2 **	64.79°	70.36°	63.71°
** Film 3 **	61.54°	79.07°	67.07°
** Average **	62.68°	75.19°	65.87°
** Standard Deviation **	1.83°	4.43°	1.88°

## Data Availability

Not applicable.

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
