# Peer review of "Development of an Oriented Co-Culture System Using 3D Scaffolds Fabricated via Non-Linear Lithography"

_materials, 2022, doi:10.3390/ma15124349_

Round 1
Reviewer 1 Report
- Show the novelty of the paper compared to the literature, however there is still much work on this topic.
- Why you choose these alloys?
- In the Introduction section, the last paragraph should contain the scientific contribution and scientific hypotheses of your research. Complete, further elaborate the scientific contribution and scientific hypotheses of your research. Be explicit. In addition to the goal of the research (which was written), the novelty in the context of the scientific contribution should be pointed out. Scientific contributions should be written based on the shortcomings of previous research in the literature. In this way, the authors will better emphasize novelty and scientific soundness.
- Analyze and discuss possibilities of practical application.
In the conclusions, state the scientific contribution, the shortcomings of your methodology and future research.
- Generally, the quality of the writing could be improved.
Author Response
Show the novelty of the paper compared to the literature, however there is still much work on this topic.
1) Why you choose these alloys?
Reply:
The reasons for choosing the materials are stated throughout the manuscript. Briefly, the most important ones are:
• The materials are well established; they are easy to prepare, handle and polymerise via MPL.
• Combined with MPL, they enable the creation of high resolution 3D scaffolds, a valuable asset in fields like Tissue Engineering. This is not always achievable by other techniques/materials.
• Based on previous studies, they are suitable for cell cultures. Furthermore, based on the contact angle measurements that were recently added, they exhibit a hydrophilic nature which would be considered beneficial for cell adhesion.
• Incorporation of the Sudan Black B dye as a photoinitiator and as a post-fabrication treatment addresses the drawback of autofluorescence that many resins used in MPL exhibit. This enables the use of fluorescence imaging techniques such as confocal microscopy.
• As already reported by Flamourakis et al (reference [17] in the manuscript), SBB post-fabrication treatment significantly smoothens the scaffold surfaces. This is expected to increase hydrophilicity of the scaffolds. In this work, this expectation is backed up by the contact angle measurements (Fig. 1C) where undeveloped polymerised films exhibited increased hydrophilicity compared to developed ones (undeveloped films should have a smoother surface since no material is removed by the solvents).
2) In the Introduction section, the last paragraph should contain the scientific contribution and scientific hypotheses of your research. Complete, further elaborate the scientific contribution and scientific hypotheses of your research. Be explicit. In addition to the goal of the research (which was written), the novelty in the context of the scientific contribution should be pointed out. Scientific contributions should be written based on the shortcomings of previous research in the literature. In this way, the authors will better emphasize novelty and scientific soundness.
Reply:
As mentioned previously, the last paragraph of the introduction was rewritten as follows, in order to include the most important findings of the study and provide a more specific overview of the aims of this work:
‘’In this work, fabrication of scaffolds with minimal autofluorescence for the cultivation of neurons and Schwann cells, both in mono- and co-culture systems for 3 different timepoints (7, 14 and 21 days) was carried out. A novel scaffold geometry was fabricated and the elongation of
the cells and neurites in a direction of choice was showcased. By using two imaging techniques (Scanning Electron Microscopy (SEM) and Confocal Microscopy), it is shown that long neurite growth (>40μm) is favoured for scaffold co-culture systems for long-term experimental periods compared to mono-culture setups where long neurites exhibit a significant decrease from 7 to 21 days. These findings highlight that, by choosing scaffolds with appropriate geometry, cell behaviour, directionality and axon growth can be influenced according to application needs. The presented results provide a first step in the creation of an in vitro model for studying neuronal cell behaviour, while also providing insight into the understanding of neurodegenerative diseases and cell processes, such as myelination, in future research.’’
Additionally, we would like to point out that paragraphs 4, 5 and 6 of the Introduction provide more insight for the aims of this study and the contribution and novelty in the field. More specifically:
• Paragraph 4 highlights the importance of co-culture system studies, since they are more limited compared to studies using a single cell type.
• Paragraph 5 highlights the advantages of MPL and the reasons it is considered a powerful technique for Tissue Engineering.
• Paragraph 6 demonstrates the shortcomings of MPL and the benefits of SBB incorporation in our scaffolds. Moreover, it should be noted that co-culture studies using the SBB dye either as a photoinitiator or as a post-fabrication step (or both) are, to the best of our knowledge, even more limited than conventional co-culture studies.
Since these paragraphs highlight the key points of our study and offer the necessary background to back up our findings, we thought that including this information in just one lengthy paragraph would be repetitive and would cause confusion. Dividing the introduction in this way would provide a better flow for the reader.
3) Analyze and discuss possibilities of practical application. In the conclusions, state the scientific contribution, the shortcomings of your methodology and future research.
Reply:
We would like to thank the reviewer for their comment. The final part of the last paragraph of the discussion was rewritten to include the following text (in red) for better clarification (lines 691-697):
‘’It must be noted that the findings in this work aim to offer an in vitro model for the study of neurodegenerative diseases and key cell responses such as myelination. However, the ultimate goal is to apply TE in real life. To this end, the possibility of using primary cells instead of cell lines would be a very interesting extension of this study. Use of primary cells could pave the way for incorporation of the presented findings in in vivo studies, such as the creation of autografts for surgery or for injectable scaffolds in the near future. Furthermore, the general concept of the presented study could also be extended to other fields of TE, such as bone tissue engineering.’’
The Conclusions were also rewritten to include the following text (in red) as suggested:
‘’In this work, an in vitro model for the study of SW10/N2a co-culture was developed. An easy-to-handle established material was used for the fabrication of scaffolds for cell culture. This study showed that by carefully adjusting scaffold topography, cell morphology and axon growth can be influenced towards specific directions. Both in mono-cultures and co-cultures of SW10 and N2a cells, it was shown that cells migrate and proliferate on the scaffolds and that they are stimulated to follow the guidewire direction compared to the random orientation observed on flat glass surfaces. It is also demonstrated that the presence of SW10 cells favours the development of longer neurites on scaffolds for long-term cultures compared to the mono-cultures, which could prove a valuable asset for practical applications. It is expected that by improving the presented system (larger scale scaffolds, further improved hydrophilicity, use of primary cells, possibility for practical applications), it could potentially serve as an even more complete tool in the fields of neurodegenerative disease study and tissue engineering.’’
4) Generally, the quality of the writing could be improved.
Reply:
Corrections suggested by all reviewers were updated in the manuscript to improve the overall quality of the manuscript.

Reviewer 2 Report
The authors present the article entitled “Development of an oriented co-culture system using 3D scaffolds fabricated via non-linear lithography”.
This paper used a novel scaffold geometry fabricated via Multiphoton Lithography (MPL), a commonly used 19 fabrication method, for the mono- and cocultures of glial Schwann (SW10) and neuronal Neuro-2a 20 (N2a) cells.
The article presents the following concerns:
Include quantitative values in the abstract in order to highlight the findings.
I suggest adding a table that summarizes the experimental setup of glial Schwann (SW10) and neuronal Neuro-2a (N2a) cells in section 2.
2.1 and 3.1 have the same name. Same in 2.2 and 3.2
Lines 318-320: How efficient is it currently?
-
The text must be written in the 3rd person or passive voice.
-
It's needed to describe the article structure in the final part of the introduction.
-
I recommend giving an introduction between sections 2 and 2.1, 3 and 3.1.
-
Justify the relevance of the publication in the journal; it is necessary to include references from the journal.
-
Figure 5 must be vectorized
The following misspelling should be checked:
-
Line 33: “Damage of tissues…” should be rewritten as “Damage to tissues…” It seems that preposition use may be incorrect here.
-
Line 131: The phrase “with respect to” may be wordy. Consider changing by “concerning”, “for” or “to”.
-
Line 212: “Sample observation and imaging was performed…” should be rewritten as “Sample observation and imaging were performed…”
-
Line 229: “the neurite with accuracy…” should be rewritten as “the neurite with an accuracy…” adding an article.
-
Line 335: “also grow axons which follow random…” should be rewritten as “also grow axons that follow random…”
-
Line 351: “mostly attach on to…” should be rewritten as “mostly attach to…”
-
Line 471: “hinderance” should be changed to “hindrance”.
Author Response
Reviewer 1
The authors present the article entitled “Development of an oriented co-culture system using 3D scaffolds fabricated via non-linear lithography”.
This paper used a novel scaffold geometry fabricated via Multiphoton Lithography (MPL), a commonly used 19 fabrication method, for the mono- and cocultures of glial Schwann (SW10) and neuronal Neuro-2a 20 (N2a) cells.
The article presents the following concerns:
1) Include quantitative values in the abstract in order to highlight the findings.
Reply:
The following text was added to include the most significant quantitative data in the abstract as suggested:
“Our findings demonstrate that scaffold co-culture systems favour the presence of neurites com-pared to mono-cultures at 21 days (31.4% ± 5.5% and 15.4% ± 5.4% respectively), while there is also a significant decrease in long neurites in the mono-culture over time (45.3% ± 15.9% at 7 days versus 15.4% ± 5.4% at 21 days).”
The information was also included in the main body of the manuscript at the end of Section 3.3.3. The following text (in red) was added to that effect:
“As far as the scaffolds are concerned, it is obvious that there is a more dramatic decrease of axons with the passing of days (Fig. 5A). For long term mono-cultures, this decrease is statistically significant when comparing 7 day and 21-day cultures (long axon percentages of 45.3% ± 15.9% and 15.4% ± 5.4% respectively, unpaired t-test: p-value=0.037) (Fig. 5A). This is not the case for the co-culture system (Fig. 5B). In the case of co-cultures, the percentage of long axons remains practically stable at all timepoints. However, when comparing the 21-day mono-cultures and co-cultures on scaffolds, it is obvious that the increase in the percentage of long neurites is statistically significant (long axon percentage of 21-day co-cultures on scaffolds: 31.4% ± 5.5%, unpaired t-test: p-value=0.0233).”
2) I suggest adding a table that summarizes the experimental setup of glial Schwann (SW10) and neuronal Neuro-2a (N2a) cells in section 2.
Reply:
Table 1 summarising the experimental setup was added in section 2 as suggested.
3) 2.1 and 3.1 have the same name. Same in 2.2 and 3.2
Reply:
An additional section on the preparation of thin films and contact angle measurements was inserted (Section 2.3). As such, Section 2.2 has been renamed to “Bridged-Shaped 3D Scaffold fabrication via MPL”, while Section 2.4 has also been renamed to: “Cell seeding and cultures on flat controls and 3D scaffolds” to avoid confusion with Sections 3.1 and 3.3 respectively.
4) Lines 318-320: How efficient is it currently?
Reply:
A qualitative interpretation of the efficiency can be drawn from the general conclusions of this work. The two key observations are:
- Neurites elongate alongside the guidewires giving specific directionality to neurites compared to flat coverslips and other regions of scaffold cultures apart from the guidewires like the surrounding area of the scaffolds or the scaffold supporting blocks.
- The scaffold co-cultures exhibit a stable number of long neurites (axons) throughout the duration of all timepoints in comparison with N2a mono-cultures. In the latter case, the number of long neurites decreases drastically.
The aforementioned points indicate that scaffolds dictate neurite directionality (in other words if a neurite is formed on a guidewire, it will follow the guidewire direction) and that the presence of SW10 cells increase the efficiency of the system towards long neurites. Comparison of the data in Figure 5 shows that for long-term cultures (21 days) on scaffolds the presence of SW10 cells gives a two-fold percentage (31.4% ± 5.5% to 15.4% ± 5.4%) of long neurites compared to the respective mono-cultures. This two-fold percentage increase may be a significant factor for practical applications since bridging the gaps of damaged tissues would require longer axons with specific directionality.
5) The text must be written in the 3rd person or passive voice.
Reply:
Several changes were made throughout the text to ensure that third person and/or passive voice were used instead.
6) It's needed to describe the article structure in the final part of the introduction.
Reply:
The final paragraph of the introduction was re-written as follows to better describe the article structure.
“In this work, fabrication of scaffolds with minimal autofluorescence for the cultivation of neurons and Schwann cells, both in mono- and co-culture systems for 3 different timepoints (7, 14 and 21 days) was carried out. A novel scaffold geometry was fabricated and the elongation of the cells and neurites in a direction of choice was showcased. By using two imaging techniques (Scanning Electron Microscopy (SEM) and Confocal Microscopy), it is shown that long neurite growth (>40μm) is favoured for scaffold co-culture systems for long-term experimental periods compared to mono-culture setups where long neurites exhibit a significant decrease from 7 to 21 days. These findings highlight that, by choosing scaffolds with appropriate geometry, cell behaviour, directionality and axon growth can be influenced according to application needs. The presented results provide a first step in the creation of an in vitro model for studying neuronal cell behaviour, while also providing insight into the understanding of neuro-degenerative diseases and cell processes, such as myelination, in future research.”
7) I recommend giving an introduction between sections 2 and 2.1, 3 and 3.1.
Reply:
We would like to thank you for your recommendation. Based on other articles published in the “Materials” journal, we do not believe that an introduction between Sections 2 and 2.1 is necessary, as the aim is to explain the methodologies employed for this work.
For section 3, the following introduction was added:
“The main findings of this work are presented in this section. 3D scaffolds were fabricated and mono-cultures of both SW10 and N2a cells, as well as co-cultures were carried out. Cell behaviour between flat controls and scaffolds was monitored in order to determine how the scaffold may be able to influence cellular parametres such as morphology, directionality, differentiation and neurite formation. To this end, both scanning electron microscopy and confocal microscopy were used.”
8) Justify the relevance of the publication in the journal; it is necessary to include references from the journal.
Reply:
Relevant references from the journal have been added to the manuscript where appropriate.
9) Figure 5 must be vectorized.
Reply:
All figures for this manuscript have been prepared using Adobe Illustrator.
10) The following misspelling should be checked:
- Line 33: “Damage of tissues…” should be rewritten as “Damage to tissues…” It seems that preposition use may be incorrect here.
- Line 131: The phrase “with respect to” may be wordy. Consider changing by “concerning”, “for” or “to”.
- Line 212: “Sample observation and imaging was performed…” should be rewritten as “Sample observation and imaging were performed…”
- Line 229: “the neurite with accuracy…” should be rewritten as “the neurite with an accuracy…” adding an article.
- Line 335: “also grow axons which follow random…” should be rewritten as “also grow axons that follow random…”
- Line 351: “mostly attach on to…” should be rewritten as “mostly attach to…”
- Line 471: “hinderance” should be changed to “hindrance”.
Reply:
All suggested corrections were made.
Reviewer 3 Report
I really appreciate your work and presentation of your results. I do not have any major comments. It is well written and the subject of study is described in detail.
Author Response
Reviewer 2
I really appreciate your work and presentation of your results. I do not have any major comments. It is well written and the subject of study is described in detail.
Reply:
We would like to thank the reviewer for their kind words.
Reviewer 4 Report
The present work proposes the development of an oriented co-culture system using 3D scaffolds fabricated via non-linear lithography The paper is fairly properly written (but the article needs a typo-grammatical check). Here are the main comments about the paper which need to be considered:
1) I would like to ask the authors to abstract clarify (in the context of specific values) the obtained research results
2) The authors in the manuscript presented scaffold visualizations. Have the authors investigated the surface roughness? It would be worth mentioning. What is more, maybe it could be measured?; I am curious what results the authors would get? E.g., by using the AFM microscope. Of course, if the authors of this type of microscope have.
Author Response
Reviewer 3
The present work proposes the development of an oriented co-culture system using 3D scaffolds fabricated via non-linear lithography The paper is fairly properly written (but the article needs a typo-grammatical check). Here are the main comments about the paper which need to be considered:
1) I would like to ask the authors to abstract clarify (in the context of specific values) the obtained research results
Reply:
As mentioned previously, the following text was added to include the most significant quantitative data in the abstract as suggested:
“Our findings demonstrate that scaffold co-culture systems favour the presence of neurites com-pared to mono-cultures at 21 days (31.4% ± 5.5% and 15.4% ± 5.4% respectively), while there is also a significant decrease in long neurites in the mono-culture over time (45.3% ± 15.9% at 7 days versus 15.4% ± 5.4% at 21 days).”
The information was also included in the main body of the manuscript at the end of Section 3.3.3. The following text (in red) was added to that effect:
“As far as the scaffolds are concerned, it is obvious that there is a more dramatic decrease of axons with the passing of days (Fig. 5A). For long term mono-cultures, this decrease is statistically significant when comparing 7 day and 21-day cultures (long axon percentages of 45.3% ± 15.9% and 15.4% ± 5.4% respectively, unpaired t-test: p-value=0.037) (Fig. 5A). This is not the case for the co-culture system (Fig. 5B). In the case of co-cultures, the percentage of long axons remains practically stable at all timepoints. However, when comparing the 21-day mono-cultures and co-cultures on scaffolds, it is obvious that the increase in the percentage of long neurites is statistically significant (long axon percentage of 21-day co-cultures on scaffolds: 31.4% ± 5.5%, unpaired t-test: p-value=0.0233).”
2) The authors in the manuscript presented scaffold visualizations. Have the authors investigated the surface roughness? It would be worth mentioning. What is more, maybe it could be measured?; I am curious what results the authors would get? E.g., by using the AFM microscope. Of course, if the authors of this type of microscope have.
Reply:
We would like to thank the reviewer for the suggestion. Unfortunately, access to equipment for roughness measurements was not available. It should however be mentioned that treatment with the Sudan Black B dye significantly smoothens the scaffold surface as shown by the SEM images of Flamourakis et al (reference [17] in the manuscript). Contact angle measurements were carried out for polymerized thin films of the resins and showed that both resins should exhibit hydrophilic characteristics (Figure 1C), with corresponding sections added in the Materials & Methods and the Results. The decision to measure the thin films was made as given the relatively small size of the fabricated scaffolds, contact angle measurements for the bridge-shaped 3D scaffolds was not possible due to their irregular shape. The fabrication of larger scaffolds, which would be more suitable for such measurements, is limited by our system since the presented scaffold geometry does not exhibit unit cell periodicity. However, we believe that the polymerized thin films should exhibit similar properties as the 3D scaffolds.
Reviewer 5 Report
The manuscript by Korda and colleagues introduces a 3D scaffold co-culture system in order to better understand the interaction between glial and neuronal cells. As a whole, it has the potential to be published after addressing the following points:
1. Since the authors investigated the effects of topography on SW10 and N2a behaviour, they should also include other surface characterization methods (e.g., AFM and contact angle measurements) that would supplement the SEM micrographs. The roughness of the substrate is correlated with surface hydrophilicity, which influences cell adhesion.
- Lines 433-434: the established cytotoxicity of the used material from the authors' previous works was based on Live/Dead assay for 48 h time point (authors' reference [14]) and has not been convincingly supported by the fluorescence images. The authors should perform more quantified characterization needed (e.g. metabolic assays) to confirm the Live/Dead results.
- Lines 473-480: I recommend the authors to interpret the results also based on the findings of a study by Angelaki et al.:
ANGELAKI, D., et al. Laser-induced topographies enable the spatial patterning of co-cultured peripheral nervous system cells. Materials Science and Engineering: C, 2020, 115: 111144.
Author Response
Reviewer 4
The manuscript by Korda and colleagues introduces a 3D scaffold co-culture system in order to better understand the interaction between glial and neuronal cells. As a whole, it has the potential to be published after addressing the following points:
1) Since the authors investigated the effects of topography on SW10 and N2a behaviour, they should also include other surface characterization methods (e.g., AFM and contact angle measurements) that would supplement the SEM micrographs. The roughness of the substrate is correlated with surface hydrophilicity, which influences cell adhesion.
Reply:
We would like to thank the reviewer for the suggestion. As mentioned previously, unfortunately, access to equipment for roughness measurements was not available. It should however be mentioned that treatment with the Sudan Black B dye significantly smoothens the scaffold surface as shown by the SEM images of Flamourakis et al (reference [17] in the manuscript). Contact angle measurements were carried out for polymerized thin films of the resins and showed that both resins should exhibit hydrophilic characteristics (Figure 1C), with corresponding sections added in the Materials & Methods and the Results. The decision to measure the thin films was made as given the relatively small size of the fabricated scaffolds, contact angle measurements for the bridge-shaped 3D scaffolds was not possible due to their irregular shape. The fabrication of larger scaffolds, which would be more suitable for such measurements, is limited by our system since the presented scaffold geometry does not exhibit unit cell periodicity. However, we believe that the polymerized thin films should exhibit similar properties as the 3D scaffolds.
2) Lines 433-434: the established cytotoxicity of the used material from the authors' previous works was based on Live/Dead assay for 48 h time point (authors' reference [14]) and has not been convincingly supported by the fluorescence images. The authors should perform more quantified characterization needed (e.g. metabolic assays) to confirm the Live/Dead results.
Reply:
As the material used in this work has already been established as cell-friendly, performing additional assays to ascertain its biocompatibility was considered redundant, since this would be outside of the scope of the presented work. The main focus of this study was not to quantify the cytotoxicity of the material but to examine how the novel geometry of the scaffold could influence cell orientation and neurite directionality. As a complementary remark, it is worth mentioning that cells exhibited scaffold saturation over time, as shown in both the SEM and the confocal images, which would further reinforce the biocompatibility of this material. Combining the knowledge from our previous work and the scaffold saturation observed in the present work was considered sufficient to highlight the suitability of the material for this work and thus, no additional assays were carried out.
3) Lines 473-480: I recommend the authors to interpret the results also based on the findings of a study by Angelaki et al.: ANGELAKI, D., et al. Laser-induced topographies enable the spatial patterning of co-cultured peripheral nervous system cells. Materials Science and Engineering: C, 2020, 115: 111144.
Reply:
A new paragraph in Section 4 was added to connect the two studies. The following text was added:
“These findings are in good agreement with previous work in our lab [24]. More specifically, Angelaki et al, have designed patterned micro-sized spikes and nano-sized ripples for the co-culture of SW10/N2a cells. There, SW10 cells showed directed adhesion on the spikes which played a pivotal role in the direction of differentiated N2a cells. Similarly to what has been presented in this work, SW10 cells saturated the scaffold surface and followed an orientation based on scaffold topography. N2a cells exhibited a similar behaviour; they showed preference for SW10-covered areas and their neu-rite directionality was influenced by SW10 orientation dramatically. In the case of flat coverslips, this orientation was random, while on the other hand, orientation of cells and directionality of neurites on scaffolds followed the scaffold geometry. The same observation could be made for N2a mono-cultures, as neurite directionality was also influenced by the scaffolds even in the absence of SW10 cells. Angelaki et al proposed that undifferentiated N2a cells do not sense topographical cues; the process of topography sensing seems to come from the neurites of differentiated N2a cells. This is also featured to an extent in the presented results, as in N2a mono-cultures (Fig. 3), differentiated N2a cells show neurite elongation that is directly affected by scaffold guidewires, whereas the same observations are made for co-cultures with the important detail that SW10 cells are always present near or under differentiated N2a cells that exhibit neu-rite directionality (Fig. 4). Key differences between these two works are the fact that in this work, the fabricated scaffolds are three-dimensional, whereas in the work of Angelaki et al, the scaffolds constitute a 2.5D environment. In addition, the starting populations of cells in the current work are far smaller, as the aim of the current study was to establish the conditions for a long-term coculturing system. As such, the lower cell numbers allowed us to observe scaffold saturation progressively over time. It is very interesting that a similar cell behaviour is present in both works, with SW10 cells seem-ing to be more sensitive to topographical cues and forming a glial carpet that supports and influences neurite directionality of differentiated N2a cells that are in contact with them. Combined, these findings show that this behaviour is encountered in both 2.5D and 3D environments and that it can be observed for different topographical characteristics.”
General remark
Due to a small proofreading error, the scale bars of merged confocal images in Figures 2, 3 and 4 were marked as 50μm, instead of the correct 30μm. The correct scale bars were fixed in the figure captions.
Round 2
Reviewer 1 Report
The presented data are original and interesting. The manuscript has been significantly improved and is suitable for publication in the present Journal.
Reviewer 5 Report
The authors have solved my concerns.